# Metabolic Fingerprinting for the Diagnosis of Clinically Similar Long COVID and Fibromyalgia Using a Portable FT-MIR Spectroscopic Combined with Chemometrics

**DOI:** 10.3390/biomedicines11102704

**Published:** 2023-10-05

**Authors:** Kevin V. Hackshaw, Siyu Yao, Haona Bao, Silvia de Lamo Castellvi, Rija Aziz, Shreya Madhav Nuguri, Lianbo Yu, Michelle M. Osuna-Diaz, W. Michael Brode, Katherine R. Sebastian, M. Monica Giusti, Luis Rodriguez-Saona

**Affiliations:** 1Department of Internal Medicine, Division of Rheumatology, Dell Medical School, The University of Texas, 1601 Trinity St., Austin, TX 78712, USA; 2Department of Food Science and Technology, The Ohio State University, Columbus, OH 43210, USA; yao.806@buckeyemail.osu.edu (S.Y.); bao.172@buckeyemail.osu.edu (H.B.); delamocastellvi.1@osu.edu (S.d.L.C.); nuguri.2@buckeymail.osu.edu (S.M.N.); giusti.6@osu.edu (M.M.G.); rodriguez-saona.1@osu.edu (L.R.-S.); 3Campus Sescelades, Departament d’Enginyeria Química, Universitat Rovira i Virgili, Av. Països Catalans 26, 43007 Tarragona, Spain; 4Department of Internal Medicine, Dell Medical School, The University of Texas, 1601 Trinity St., Austin, TX 78712, USA; rija.aziz@austin.utexas.edu (R.A.); michelle.osuna@austin.utexas.edu (M.M.O.-D.); william.brode@austin.utexas.edu (W.M.B.); kate.sebastian@austin.utexas.edu (K.R.S.); 5Center of Biostatistics and Bioinformatics, The Ohio State University, Columbus, OH 43210, USA; lianbo.yu@osumc.edu

**Keywords:** fibromyalgia, post-acute sequelae of SARS-CoV-2 (PASC)/long COVID, metabolic fingerprinting, in-clinic disease diagnostics, chemometrics, blood

## Abstract

Post Acute Sequelae of SARS-CoV-2 infection (PASC or Long COVID) is characterized by lingering symptomatology post-initial COVID-19 illness that is often debilitating. It is seen in up to 30–40% of individuals post-infection. Patients with Long COVID (LC) suffer from dysautonomia, malaise, fatigue, and pain, amongst a multitude of other symptoms. Fibromyalgia (FM) is a chronic musculoskeletal pain disorder that often leads to functional disability and severe impairment of quality of life. LC and FM share several clinical features, including pain that often makes them indistinguishable. The aim of this study is to develop a metabolic fingerprinting approach using portable Fourier-transform mid-infrared (FT-MIR) spectroscopic techniques to diagnose clinically similar LC and FM. Blood samples were obtained from LC (*n* = 50) and FM (*n* = 50) patients and stored on conventional bloodspot protein saver cards. A semi-permeable membrane filtration approach was used to extract the blood samples, and spectral data were collected using a portable FT-MIR spectrometer. Through the deconvolution analysis of the spectral data, a distinct spectral marker at 1565 cm^−1^ was identified based on a statistically significant analysis, only present in FM patients. This IR band has been linked to the presence of side chains of glutamate. An OPLS-DA algorithm created using the spectral region 1500 to 1700 cm^−1^ enabled the classification of the spectra into their corresponding classes (Rcv > 0.96) with 100% accuracy and specificity. This high-throughput approach allows unique metabolic signatures associated with LC and FM to be identified, allowing these conditions to be distinguished and implemented for in-clinic diagnostics, which is crucial to guide future therapeutic approaches.

## 1. Introduction

The Post Acute Sequelae of SARS-CoV-2 infection (PASC), also known as Long COVID (LC), is characterized by lingering symptomatology following the initial COVID-19 illness and is often debilitating. As described by the National Institute for Health and Care Excellence (NICE) guidelines, it is defined as signs and symptoms that develop during or after an acute infection consistent with COVID-19 that persists longer than 4 weeks [1]. More than 200 PASC/LC symptoms have been reported, which can affect multiple organ systems and last weeks, months, or years [2,3,4,5,6,7]. Although there are no consensus diagnostic criteria, generally, PASC is organized into two domains. The first is tissue or organ injury identifiable through traditional laboratory or imaging technologies, usually following severe acute COVID-19 illness and hospitalization, and the new onset of a major disease like diabetes or myocardial infarction following the infection. These are typically called post-COVID conditions. By contrast, LC is generally understood as a group of symptoms persisting for more than 6 months that have significant functional impairment, usually in the absence of identifiable major organ injuries. [1,8,9]. Patients with LC often suffer from widespread pain, fatigue, post-exertional malaise, physical and cognitive dysfunction, psychological factors, sleep disturbance, and autonomic dysregulation, amongst many others [7,8,9,10,11]; LC can be seen in up to 10–30% of individuals following infection and has also been recognized to develop in a small portion of individuals following SARS-CoV-2 vaccination [7]. With over 65 million individuals affected by LC worldwide, as a number that continues to grow, the treatment of LC is a global health crisis. Moreover, the mechanism of chronic pain in LC has not been definitively defined [12,13].

Fibromyalgia (FM) is a chronic musculoskeletal pain disorder that often leads to functional disability and may progress to a severe impairment of quality of life, affecting up to 5% of individuals worldwide [14,15,16,17]. As the etiopathogenesis of FM remains poorly understood, there is currently no biomarker, objective, or reliable test available for diagnosing FM. The type of pain experienced by individuals with FM is chronic non-malignant pain; it is essentially identical to the type of pain experienced by patients with LC. Despite this common clinical phenotype, the underlying pathogenesis of these disorders varies greatly, and particularly for FM, is not well understood. In addition, these disorders may frequently overlap, making diagnosis even more challenging. Therefore, the absence of distinct diagnostic markers and similar clinical presentations in these conditions creates a conundrum in terms of the ability to accurately differentiate and diagnose LC and FM [18,19,20].

LC and FM patients with poorly explained symptoms are often treated as individuals with widespread pain and are subsequently inappropriately treated with narcotics. An increase in opioid prescriptions for chronic non-malignant pain-related syndromes has been seen and exacerbated in the United States since the onset of the COVID-19 pandemic [21,22,23,24,25,26,27]. This is vitally important to recognize since a large percentage of patients with chronic non-malignant pain regularly seek and obtain analgesics for the treatment of their global pain complaints. However, there is no evidence that opioids improve their status beyond standard care and may contribute to a less favorable outcome [28,29]. A prior survey of chronic pain patients has shown that 49% of patients taking opioids continued to report severe pain (>7/10), and 41% of those surveyed meet current criteria for FM [30]. Although this survey preceded the identification of LC, it is likely that similar numbers would be seen in a LC cohort since the clinical characteristics and level of pain seen in this group are comparable to those seen in FM and CFS/ME [10]. Over-reliance on opioids for chronic pain disorders may be due to the complexity of managing chronic pain conditions [31,32,33].

Metabolic fingerprinting for accurate diagnosis is crucial to guide treatment approaches, utilize reliable biomarkers for the differentiation of clinically similar diseases, the identification of at-risk populations, and the ability to administer targeted and appropriate medication for specific treatment groups, all of which are urgently needed. Furthermore, the elucidation of specific biomarkers holds promise for providing important clues for the development of personalized treatment approaches in the future of these conditions [34,35]. Common analytic techniques for profiling metabolic fingerprints include nuclear magnetic resonance (NMR) and mass spectrometry (MS) with enough selectivity and specificity [36]. However, these techniques are less amenable to being implemented in clinics due to the demands of costly instrumentation, tedious sample preparation, and well-trained technicians [37]. Technique breakthroughs of vibrational spectroscopic techniques, specifically FT-IR combined with chemometrics, have provided new opportunities to explore the metabolic fingerprinting of diseases [34,38,39,40]. Chemometric analyses such as principal component analysis (PCA), the soft independent modeling of class analogy (SIMCA), partial least squares—discriminant analysis (PLS-DA), and support vector machine (SVM) are commonly used to extract spectral fingerprints from other chemical properties of biological samples [37]. In addition, since the last decade, portable FT-IR spectrometers have become commercially available, advancing in optoelectronics, micro-electro-mechanical systems, and (MEMS) production [41]. The applications of portable FT-MIR spectrometers with high spectral resolution equivalent to benchtop instrument systems have the tremendous potential to profile metabolic fingerprints, enabling in-clinic diagnosis.

Our group reported the first metabolomics studies to diagnose FM and related rheumatologic disorders (rheumatoid arthritis (RA), systemic lupus erythematosus (SLE), and osteoarthritis (OA) using vibrational spectroscopy [37,42,43]. A semi-permeable membrane filtration extraction approach was standardized to isolate the low molecular weight fraction. The significance of aromatic amino acids and peptide backbones was highlighted, and aromatic amino acids might serve as candidate biomarkers for FM [37,42]. Since 2020, some research has been conducted on the rapid diagnosis of COVID-19 using FT-IR spectroscopy, such as detecting the virus in blood samples [44], saliva [45,46], and pharyngeal cell smears [47]. Lipids, proteins, and nucleic acids are strongly associated with COVID-19-positive samples, serving as “chemical fingerprints” [46,47]. Dierckx et al. employed NMR to evaluate the association of blood metabolites with disease severity in COVID-19 patients and reported that a broad set of biomarkers, including amino acid concentration and inflammatory markers, were correlated with disease severity [48]. However, to date, metabolic fingerprinting for the diagnosis of clinically similar LC and FM has not been investigated and is, therefore, the objective of this study. In addition, unique metabolic fingerprinting signatures associated with LC and FM were assessed through spectral deconvolution, providing unique information for discriminating LC and FM.

## 2. Materials and Methods

### 2.1. Patient Sample Recruitment and Sample Storage

Approval from the University of Texas at the Austin institutional review board was obtained prior to embarking on any study on human subjects. All studies adhered to the Declaration of Helsinki principles. The IRB approval date was (study no. 2020030008) 19 June 2020. Following informed consent, blood samples were obtained from patients with LC (*n* = 50), FM (*n* = 50), and healthy controls (NS, *n* = 6) at the University of Texas at the Austin Post COVID Program and Fibromyalgia and Central Sensitivity Syndrome clinics located at University Texas Health Austin Clinics, Austin, Texas. Bloodspots on LC subjects were obtained between November 2022 to March 2023. Bloodspots on patients with FM and healthy controls (HC) were obtained from September 2020 through March 2023. Samples were collected and stored on bloodspot cards (Whatman 903Blood Protein Saver Snap Apart Card, GE Healthcare, Westborough, MA, USA) at −20 °C until they were shipped to the Rodriguez-Saona Vibrational Spectroscopy laboratory at The Ohio State University Department of Food Sciences on dry ice and stored for subsequent analysis. Standardized circles on the filter paper served as a guide to ensure the collection of approximately 50 µL of blood per spot.

Questionnaires: All subjects with FM provided a self-report of symptoms through the use of the Revised Fibromyalgia Impact Questionnaire (FIQR), a 10-item self-rating instrument that measures physical functioning, work status, depression, anxiety, sleep, pain, stiffness, fatigue, and wellbeing [49]. The Beck Depression Inventory (BDI) is a 21-item self-report rating inventory that measures characteristic attitudes and symptoms of depression [50]. The Symptom Impact Questionnaire Revised (SIQR) is the FM-neutral version of the FIQR and does not assume that the patient has FM [51]. The SIQR was utilized as a measure of physical functioning, work status, depression, anxiety, sleep, pain, stiffness, fatigue, and wellbeing in all subjects with LC. All subjects with LC completed the Fibromyalgia rapid screening tool (FIRST) questionnaire. The FIRST questionnaire is a 6-item questionnaire that is used as a tool to detect the symptoms indicative of Fibromyalgia. A cut-off score of 5 (corresponding to the number of positive items) gives the highest rate of correct identification in patients (87.9%), with a sensitivity of 90.5% and a specificity of 85.7% [52].

Criteria for the diagnosis of FM included: age 18–80 with a history of FM and meeting current criteria for FM. [15] To become a patient in the Post-COVID-19 Program, individuals were required to have reached a minimum of 12 weeks from the onset of their initial COVID-19 illness. Documentation of a positive COVID-19 test was not required to receive services, although all patients were screened by a dedicated nurse to validate that their clinical history was consistent with a prior COVID-19 infection and that their symptoms are likely associated with PASC and not an obvious alternative etiology. Sigmaplot v15.0 and SigmaStat v4.0 software (Inpixon, Palo Alto, CA, USA) were utilized for the statistical analysis of questionnaires.

### 2.2. Sample Preparation

Sample preparation was conducted following a washed semi-permeable membrane ultrafiltration extraction approach with minor modifications [37]. One circle from the bloodspot cards was punched and extracted with 1 mL of HPLC-grade water in a 15 mL centrifuge tube. The mixture was sonicated (Sonic Dismembrator Model 100, Fisher, Pittsburgh, PA, USA) for 30 min to ensure thorough mixing, and the dissolved blood aliquots were subjected to the filtration process. The Amicon^®^ Ultra-4 centrifugal filter tubes (10 MWCO KDa) underwent a thorough washing process to eliminate the glycerol that coated the filter membrane. Each filter tube was washed four times with 3 mL of HPLC grade water via centrifugation (Sorvall™ Legend™ XFR Centrifuge, Thermo, Waltham, MA, USA) at 4000 rpm for 10 min at a temperature of 4 °C. Then, the dissolved blood aliquot was transferred to the washed Amicon^®^ filter tube and centrifuged at 4000 rpm for 15 min at 4 °C to remove proteins and the isolated low-molecular-weight-fraction (LMF) of the human plasma proteome. Blood filtrate fluid with LMF as a significant source to identify plasma-based biomarkers of disease [53] was dried on a film via vacuum centrifuging (Vacufuge plus Concentrator, Eppendorf, Westbury, NY, USA).

### 2.3. Spectral Data Acquisition

Spectral data were collected using a 4500a series Agilent’s portable FT-MIR spectrometer (Agilent Technologies, Santa Clara, CA, USA). This FT-MIR spectrometer was equipped with a 3-bounce diamond attenuated total reflectance (ATR) crystal, a Michelson interferometer, a zinc selenide beam splitter, and a thermoelectrically cooled dTGS detector, enabling analysis across the spectral range from 4000 to 700 cm^−1^. The ATR crystal featured a 200 µm active area on a 2 mm diameter sampling surface, providing a penetration depth of approximately 6 µm. For spectral collection, a dried blood plasma pellet was redissolved in 10 µL of HPLC-grade water. The redissolved plasma was vortexed to mix thoroughly, and 2 µL of it was pipetted onto the ATR sampling window for spectral acquisition. The excess water was evaporated using the vacuum to obtain a film on the ATR sampling window to avoid the interference of signals from water. A background was obtained between each reading, and 128 scans were co-added with an 8 cm^−1^ resolution for spectral collection to enhance the signal-to-noise ratio.

### 2.4. Chemometrics Analysis

Chemometrics analysis was used to analyze IR spectral differences, resolve unique metabolic fingerprints, and classify spectra according to their assigned classes (FM and LC). The spectra were imported into a chemometrics analysis software, Pirouette^®^ version 4.5 (Infometrix Inc., Woodville, WA, USA), to perform Orthogonal Signal Correction-Partial Least Squares Discriminant Analysis (OPLS-DA). OPLS-DA is a supervised learning technique that calculates a regression relationship between IR fingerprinting data and a response variable that contains known class memberships of FM (class 1) and LC (class 2). Orthogonal signal correction (OSC), a data filtering technique, operates by identifying and removing signal components that are orthogonal or unrelated to the response variable of interest and minimizes interindividual variance [54]. PLS-DA extracts a sequence of factors/latent variables, maximizing the covariance from both X and Y and reducing a large number of variables. Overall, by leveraging OPLS-DA, IR fingerprinting data can be effectively utilized to distinguish and classify samples belonging to FM and LC classes.

Spectral data were split into two sets randomly to train and external-validate a predictive algorithm to diagnose FM and LC. The training set contained 80% of data with FM (*n* = 40) and LC (*n* = 40). The remaining 20% of data formed an independent external validation set with FM (*n* = 10) and LC (*n* = 10). Spectral data were preprocessed to minimize undesired variation or technical artifacts from raw data, improving the quality of spectral data [55]. First, raw spectral data were normalized and smoothed using the Savitzky–Golay filter (SG, 5-points), and second derivative-transformed (SG, 7-points). This pretreatment effectively enhanced minor bands, resolved overlapping bands, and suppressed undesirable spectral features (i.e., scattering effects). Additionally, mean centering was performed to alleviate multicollinearity [56]. The combination of SG filtering and mean centering ensured that the spectral data were optimized for subsequent analysis.

Internal cross-validation (ICV) and external validation (EV) were used to assess the discriminating ability of the OPLS-DA model [57]. ICV was performed using a leave-one-out method, where each sample was excluded in turn to generate a model that could predict class membership using the remaining samples [58]. The ICV provided diagnostic statistics, such as misclassification and Rval, indicating the performance of the training model [59].

The results obtained from cross-validated OPLS-DA represent the classification of samples within the training set. This approach assisted in selecting the optimal number of latent variables (LVs) and provided insights into the classification performance of the OPLS-DA model. For EV, an independent external validation set (20%) was employed. This set was unseen by the training model and served to evaluate the model’s performance in an unbiased manner. The EV assessment provided measures of predictive accuracy, sensitivity, and specificity, resembling real-world applications in a clinical setting. Overall, the performance of OPLS-DA models was evaluated using the standard error of cross-validation (SECV), standard error of prediction (SEP), coefficient of determination (R^2^), sensitivity, specificity, and accuracy. Additionally, a receiver operating characteristic (ROC) curve was performed in R software [60] using the pROC package [61] from ICV and EV.

### 2.5. Spectra Deconvolution Analysis

The relative percentage area of each IR band in the region of 1500 to 1700 cm^−1^ was determined by the second-derivative transformation of raw absorption spectra to resolve overlapping bands using OriginPro 2023 (Origin Lab, MA, Northampton, USA) [62]. Raw spectral data were first normalized, smoothed (SG, 5-points), second derivative-transformed (SG, 7-points), multiplied by (−1) to invert second derivative IR bands, and finally baseline-corrected. Then, absorbance spectra were fitted using full width at half maximum (FWHM) Gaussian band profiles using the Multipeak option (Figure 1). In the fitting process, the height and width of selected IR bands varied until the best fit with the experimental curve (Chi-square tolerance value of 1 × 10^−9^, <400 iterations) was found. The goodness of the fit was determined by observing the F-statistics and values of Chi-square [63]. In order to test the statistical significance of the relative percentage area of each IR band deconvoluted, one-factor analysis of variance (ANOVA) was performed using OriginPro 2023 (Origin Lab, Northampton, MA, USA). The mean values were compared using Tukey’s test at the 5% significance level.

## 3. Results

### 3.1. Clinical Characteristics of Subjects 

The clinical characteristics of patients with LC and FM are presented in Table 1. Table 1 shows that subjects with LC (*n* = 50, F: 32, M: 18) had a mean age of 49.5 ± 14.6 with a range of 18–73. Their BMI was 29.5 ± 28.6, with a mean SIQR of 44.6 ± 21.4. Patients with FM (*n* = 50, F: 50, M: 0) had a mean age of 45.0 ± 13.1 with a range of 18–72. Their BMI was 31.2 ± 8.2, with a mean FIQR of 50.96 ± 22.0 and a mean BDI of 18.31 ± 9.9. Further analysis of the LC group comparing males and females is shown in Table 2. There was no statistically significant difference between the male and female LC subjects. Six control samples (HC) were utilized as a further comparator. These subjects were on no medications and did not have LC or FM. The Shapiro–Wilk normality test was conducted on the LC subjects, and the test was not significant (*p* = 0.319), indicating the normal distribution of the samples. Table 3 shows a comparison of *p* values between FIQR/SIQR, age, and BMI between the full group LC and FM subjects. There was no statistically significant difference in these parameters between the groups.

Table 4 provides a comprehensive view of all medications that subjects reported taking prior to sample collection. The most frequently used medication in both groups was gabapentin (12-LC, 13-FM). Tricyclic antidepressants were more prevalent in FM patients than LC (LC: 3, FM: 15). Twelve subjects were taking opioid analgesics (LC: 5, FM: 7), ten subjects were on selective serotonin reuptake inhibitors (SSRI) (LC: 6, FM: 4) and six subjects reported taking Naltrexone (LC: 5, FM: 1; naloxone dosage between 1–4.5 mg).

### 3.2. IR Spectroscopy

Figure 2 shows a representative spectrum of extracted LMF from the blood of FM and LC patients. The spectral profiles of all samples in both classes were similar through visual inspection. A broad and strong band around 3600–3100 cm^−1^ was attributed to O–H stretching, and the bands at 2800–3000 cm^−1^ were associated with the C–H stretching of hydrocarbon chains [58]. The weak band at 2960 cm^−1^ and the peak at 2920 cm^−1^ were attributed to the asymmetric stretching of sp_3_ and sp_2_ hybridized C–H groups, respectively, while the band centered at 2850 cm^−1^ was associated with symmetric methyl and methylene C–H stretching [64,65]. The band centered at 1740 cm^−1^ was related to C=O stretching, probably from the ester linkage of lipids and cholesterol [66,67,68]. A sharp band at 1550 cm^−1^ with a slight shoulder at 1670 cm^−1^ were associated with amide II and amide I (C=O stretch) conformations of the peptide backbone [37,67]. C–H umbrella deformations [69] and the hyperconjugation effect on methyl bending modes contributed to the peak around 1400 cm^−1^ [68,70]. Vibrations of phosphodiester bonds in RNA and phospholipids were related to the bands at 1245 cm^−1^ and 1088 cm^−1^ [68]. Additionally, the bands around 1000 cm^−1^ was associated with C–O–C stretching from cholesterol, phospholipids, and triglycerides [68] and C–O stretching from nucleic acids and polysaccharides [67].

Figure 3 shows a representative curve-fitted inverse second derivative for FM and LC patients after the deconvolution procedure in the region from 1500 to 1700 cm^−1^. Four IR bands were detected for FM patients in this spectral region, 1565, 1588, 1639, and 1670 cm^−1^, and only three IR bands were detected for LC patients, 1581, 1635, and 1670 cm^−1^. A unique spectral biomarker at 1565 cm^−1^ was identified in FM patients using the deconvolution of spectral data. This peak has been linked to stretching vibrations of carboxylate groups of amino acid side chains and, more specifically, to the presence of glutamate [71,72].

The relative percentage area of these deconvoluted IR bands is shown in Table 5. The IR bands at 1639 and 1670 cm^−1^ were linked to β-sheets and α-sheets of amide I (C=O stretch) [72] conformations of peptide backbone and were not statistically different between diseases with an average relative percentage area close to 4.7 and 21.3%, respectively. Nonetheless, the average of the relative percentage area of the IR band at 1580 cm^−1^ was associated with the functional group –C=N and stretching vibrations of carboxylate side chains [42] and was statistically significant between diseases and significantly higher for LC patients (70.3% vs. 38.2%).

The spectra of the six healthy subjects were also deconvoluted in the 1500 to 1700 cm^−1^ region, and an example is shown in Figure 4. The spectral biomarker at 1565 cm^−1^ detected in FM patients was not present. Nonetheless, another IR band at 1545 cm^−1^ not detected in FM and LC patients was clearly identified. This peak has been linked to C–N stretching and N–H bending vibrations of amide II or deprotonated carboxylate groups of peptides. The average relative percentage area of this band was close to 4.1% [71,72,73].

OPLS-DA was used to visualize the separation of LC and FM using pre-treated data. In OPLS-DA analysis, a dummy Y matrix (variable vector) consisting of class 1 (FM) and class 2 (LC) was correlated with the X matrix (spectral data [74]. In our study, the best OPLS-DA model with 80% of the samples for LC and FM diagnosis was obtained using the spectral region 1528 to 1624 cm^−1^, and the distinct spectral biomarker identified for FM patients was located using spectral deconvolution. Raw spectra were pre-processed (normalization, smoothing SG, 5 points), and the second derivative (SG, 7 points) was applied with the use of one orthogonal signal correction (OSC) component, which made the predictive quality of the model satisfactory.

The OPLS-DA model showed figures of merit (R and SECV/SEP) that were compatible with high performance (Table 6) and low *p*-value (*p* < 0.05), suggesting significant discrimination and significant differences between LC and FM patients. The score plot of OPLS-DA regression models obtained from spectral data is presented in Figure 5a. The score plots showed distinctive clusters of spectra from subjects with FM and LC. The first LV for both FM and LC classes explained 75.9% of the variance and provided an excellent regression coefficient of cross-validation (Rcv) at 0.98 (Table 6). One latent variable was sufficient since it already correctly classified all samples with a low SECV value (0.10) and no misclassification for the leave-one-out model.

Regression vector (Figure 5b) analysis showed that the discriminating region was dominated by the bands centered at 1560 and 1579 cm^−1^. By narrowing the spectral range to the spectral region where the biomarker was located, systematic variation unrelated to the disease was removed. Notably, our OPLS-DA model was able to correctly predict if the patients suffered from LC and FM in 20 out of 20 of the samples used for the EV set, with only one latent variable achieving 100% specificity, sensitivity, and accuracy (Table 6).

ROC was used to assess the performance of our diagnostic test, evaluating all possible thresholds that determined the result as positive [75]. The area under the ROC curve summarizes the predictive accuracy of the model, where a value closer to one indicates good accuracy [75]. For both cross-validated and external-validated samples, excellent prediction performances were obtained with an AUC of one, reflecting superior accuracy (Figure 6).

## 4. Discussion

The findings of this pilot study are intriguing, although definitive conclusions are limited by the size of the cohort. The clinical groups were generally similar in terms of age and BMI. The FIQR is a validated surrogate marker of pain in subjects with FM. Similarly, the SIQR is a FM-neutral questionnaire that asks identical questions but does not assume that patients have FM. Patients with LC present with a wide array of clinical symptoms, many of which mirror the symptoms and complaints of patients with FM. The values in both groups were statistically similar, suggesting that there were similar levels of pain experienced in the LC and FM groups. The FM group was 100% female, while the LC group had 18 males and 32 females. An obvious limitation regards the statistical power and generalizability of our data due to the size of the cohort, gender differences, and medication usage, amongst others. The inclusion of a 100% female FM group versus a mixed-gender LC group presents potential biases and lacks gender representation, impacting the reliability of these results. However, we found there was no statistically significant difference between the male and female LC subjects with regard to age, BMI, or SIQR, as noted in Table 2. Furthermore, normality tests suggest that these data have a normal distribution, which strongly suggests its potential for reproducibility. Medications of recruited patients were recorded at the time of blood collection. This pilot study was not powered to determine the effect of medications; however, spectroscopy data showed that there was no obvious signal/influence from medications. With the relatively small *n* of 50 in each cohort, we are not able to definitively eliminate medications as a confounding variable. There was a general similarity with regard to medication usage between the LC and FM cohort, particularly with regard to the use of gabapentin (12-LC, 13-FM), opioid analgesics (LC: 5, FM: 7), and selective serotonin reuptake inhibitors (SSRI) (LC: 6, FM: 4). There were some differences noted with tricyclic antidepressants being more prevalent in FM subjects than LC subjects (LC: 3, FM: 15) and naltrexone use was noted more in LC subjects [LC: 5, FM: 1; naltrexone dosage between 1–4.5 mg]. Whether these changes could have affected our spectral characteristics is unknown at this time and requires studies with larger cohorts for further analysis, which we are currently undertaking. Future studies should also seek to mitigate possible medication effects in the analysis. We aim to record patient medication usage, categorize the medication by type, and quantify it by the amount of usage. We can associate medication usage with each metabolite biomarker through correlations or logistic regression models. In addition, with a larger cohort, we can compare subjects on similar medications, such as gabapentin, while eliminating confounder medications from the analyses.

FT-MIR spectroscopy has been proposed as a feasible solution to investigate specific biomarkers [76,77]. The deconvolution analysis of spectral data identified a unique spectral band at 1565 cm^−1^ that was only present in FM patients. Selecting this specific spectral region to build up the OPLS-DA algorithm allowed us to successfully develop the OPLS-DA predictive algorithm to effectively distinguish between individuals with FM, LC, and those who were healthy. This discriminating IR band was associated with the presence of carboxylic amino acids, such as Glu and Asp. Glu/Asp side chain residues have a distinct symmetric vibration located around 1556–1568 cm^−1^ [78]. Glutamic acid decarboxylase (GAD) is a rate-limiting enzyme in the conversion of glutamate to gamma-aminobutyric acid. We hypothesize that FM subjects have a decreased GAD expression or activity, leading to symptoms that, in turn, appear to further decrease GAD expression and/or activity [72]. It has been reported that this cycle is not easily interrupted by behavioral or pharmacological interventions. The importance of glutamate has been speculated to favor the dysregulation of pain processing in the central nervous system of FM patients, which is particularly associated with an increase in cerebral glutamate levels. Furthermore, there is evidence to support an association between increased glutamate levels and an increase in FM symptoms [79]. In clinical studies, an increase in glutamate levels has been observed in the brains of FM patients [80,81]. Furthermore, pregabalin has been observed to reduce glutamatergic activity in the insula, and some subgroups of patients with FM have responded to treatment with N-methyl-D-aspartate (NMDA) glutamate receptor antagonists, suggesting an increase in glutamatergic activity. Parallel to this clinical evidence, animal studies in the non-inflammatory pain model have shown increased glutamate release in their spinal and ventromedial rostral cords [82]. Moreover, recent studies have shown higher serum concentrations of several amino acids such as glutamate, glycine, isoleucine, leucine, methionine, ornithine, phenylalanine, sarcosine, serine, taurine, tyrosine, and valine in FMS patients when compared with healthy controls [83]. Our healthy subjects did not have a distinct IR band at 1565 cm^−1^, which is potentially linked to the presence of Glu in sufficient concentrations in the filtered blood samples to detect its unique spectral signature. Nonetheless, another IR band detected at 1545 cm^−1^ could be key to discriminating between LC and healthy patients.

Our pilot results must be interpreted with caution lest we run the risk of overgeneralization. First of all, although our findings are intriguing, we will exercise caution rather than extrapolate our results to be indicative for all individuals affected with LC since this study is not powered to evaluate whether these signatures are characteristic of all affected with LC or only a subset of specific variant. Indeed, it is unclear at this time whether all LCs are the same or if it varies in severity between variants. Secondly, enrollment bias could exist due to differences in male/female ratios in groups and medication differences between groups amongst others. Future studies with much larger sample sizes should mitigate these types of concerns.

## 5. Conclusions

The use of a portable FT-MIR spectrometer combined with the chemometric OPLS-DA technique proved to be a promising tool for screening/diagnosis and the distinction of individuals with LC from FM with simple operational procedures and providing excellent accuracy, sensitivity and specificity. Additionally, the deconvolution of spectral data in the 1500 to 1700 cm^−1^ region, allowed us to identify a unique biomarker for FM and for NS patients that need to be further studied with metabolomics analyses via LC-MS/MS. Future studies with a larger number of samples need to be performed to further prove the findings of our study.

## Figures and Tables

**Figure 1 biomedicines-11-02704-f001:**
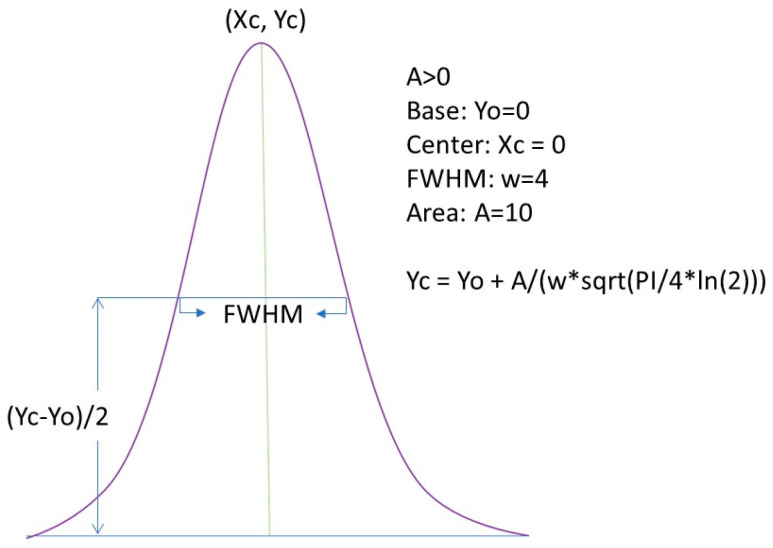
Full width at half maximum (FWHM) Gaussian fit applied to calculate %Area of each IR band detected after the pretreatment of the spectral data. Adapted from OriginPro 2023 manual.

**Figure 2 biomedicines-11-02704-f002:**
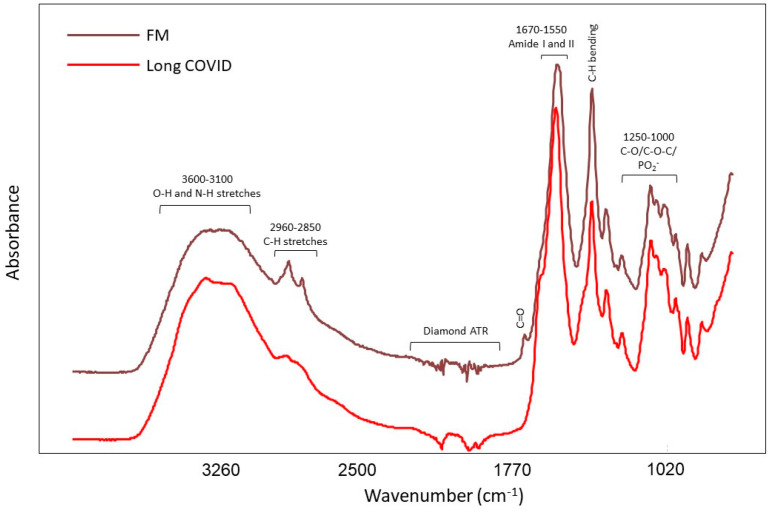
Representative IR spectrum of extracted LMF from the blood of FM and LC patients.

**Figure 3 biomedicines-11-02704-f003:**
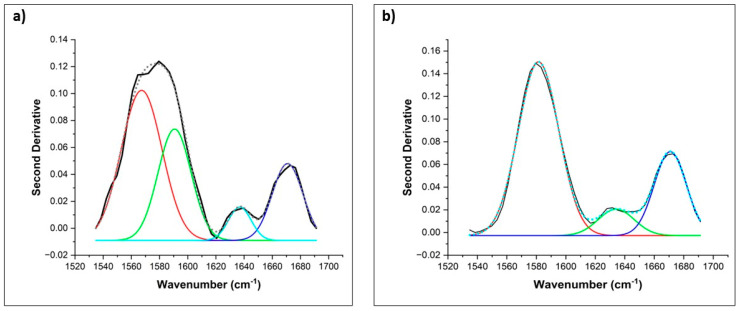
Example of curve-fitted and inverted second derivative of the 1500 to 1700 cm^−1^ region (**a**) and the IR bands deconvoluted show four IR bands (1565 cm^−1^, red band; 1588 cm^−1^, green band; 1639 cm^−1^, turquoise band; and 1670 cm^−1^, marine band) for FM patients and three IR bands (1581 cm^−1^, red band; 1635 cm^−1^, green band and 1670 cm^−1^, marine band) for LC patients (**b**) obtained with full width at a half maximum Gaussian fit. (–) Original spectral data, (…) Fitted area under the curve.

**Figure 4 biomedicines-11-02704-f004:**
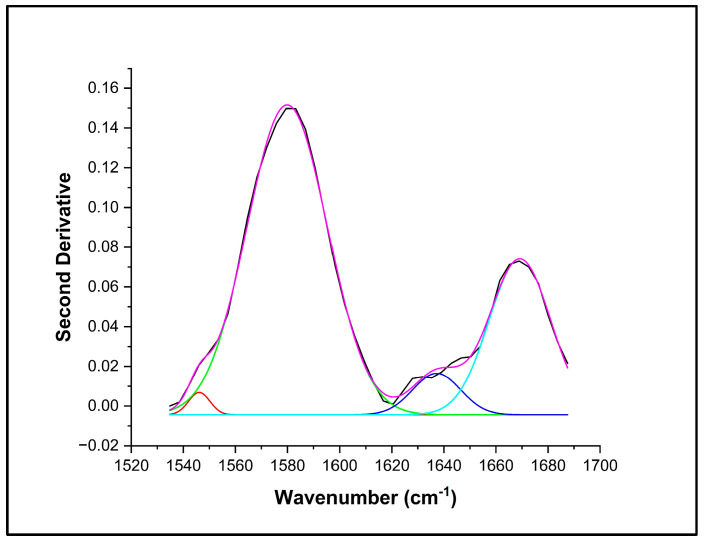
Example of curve-fitted-inverted second derivative of the 1500 to 1700 cm^−1^ region and the IR bands deconvoluted showing four IR bands (1545 cm^−1^, red band; 1580 cm^−1^; pink band; 1639 cm^−1^, marine band and 1670 cm^−1^, turquoise band) obtained with a full width at half maximum Gaussian fit for NS patients. (–) Original spectral data, (…) Fitted area under the curve.

**Figure 5 biomedicines-11-02704-f005:**
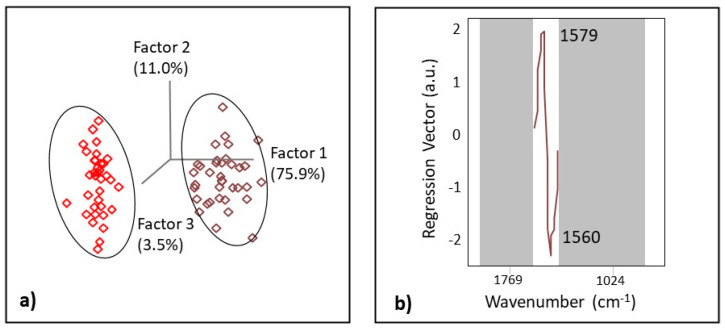
The score plot (**a**) with the first three latent variables (LVs) and the regression vector (**b**) of the OPLS-DA model obtained from spectral data of FM and LC patients.

**Figure 6 biomedicines-11-02704-f006:**
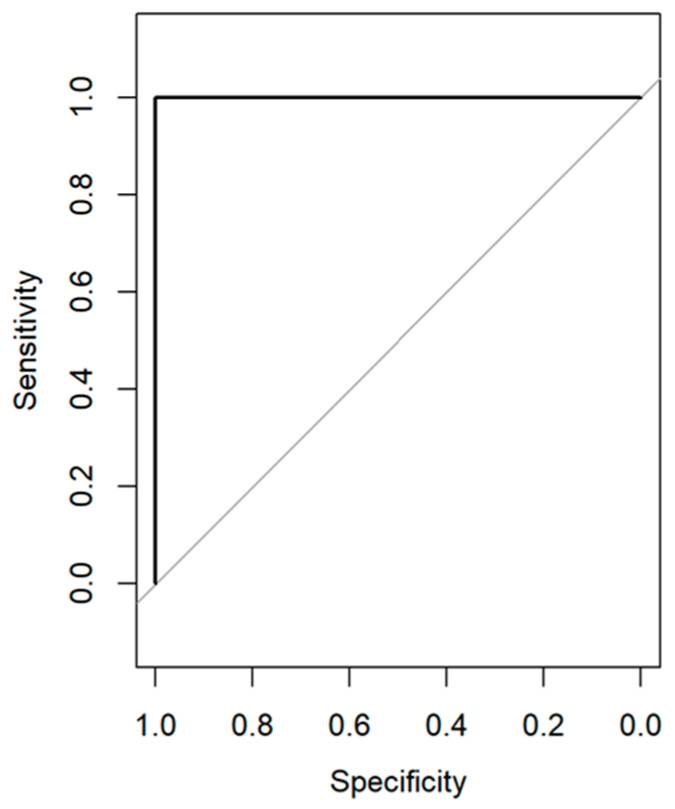
ROC diagnostic curve for cross-validated (black) and external-validated (black, overlapped with cross-validated) samples.

**Table 1 biomedicines-11-02704-t001:** Clinical characteristics of all subjects. Values expressed as mean ± /sd; N = number of subjects, age (range). LC: Long COVID; FM: Fibromyalgia, BMI: body mass index. FIQR: Fibromyalgia impact questionnaire revised. SIQR: Symptom Impact Questionnaire Revised. BDI: Beck Depression Index. FIRST: Fibromyalgia rapid screening tool HC: Healthy control values reflect number of LC subjects with values of 5 or greater.

	Age	N(M/F) [%M/%F]	BMI	FIRST	SIQR	FIQR	BDI
**LC**	49.5 ± 14.59	50 (18/32) [36/64]	29.47 ± 8.3	18/50	44.63 ± 21.4		
**FM**	44.9.0 ± 12.8	50 (0/50) [0/100]	31.57 ± 8.0			49.52 ± 21.5	18.31 ± 9.9
**HC**	45.8 ± 19.1	6(4/2) [67/33]	25.3 ± 2.4		13.95 ± 27.0		0.5 ± 0.5

**Table 2 biomedicines-11-02704-t002:** Sub-analysis of Long COVID subjects. Values expressed as mean ± sd. *p* values represent one tailed comparison between males and females for age, BMI and SIQR. The Shapiro–Wilk normality test passed.

LC	Age	BMI	SIQR
**Male**	51.8 ± 11.6	28.9 ± 4.8	41.1 ± 15.9
**Female**	48.6 ± 16.0	29.6 ± 10.0	44.8 ± 23.0
***p* value**	0.232	0.389	0.301
**Shapiro–Wilk normality test**	*p* = 0.319		

**Table 3 biomedicines-11-02704-t003:** Comparison of FIQR/SIQR, age, and BMI between FM and LC subjects. *p* values represent one tailed comparison between groups.

	FIQR/SIQR	Age	BMI
***p* value**	0.158	0.055	0.140

**Table 4 biomedicines-11-02704-t004:** Medication log for subjects.

	Medications ^a^			Medications ^a^			Medications ^a^	
Subject No.	LC	FM	Subject No.	LC	FM	Subject No.	LC	FM
1	1, 3, 8, 10, 12	2, 4, 17, 18, 21	21	2, 7, 8, 10, 15	23, 26, 31	41	1	1
2	2, 5, 22	10, 26	22	2, 6, 13	4	42	2, 14, 15, 26	3, 18, 20, 21
3	8, 14, 23, 24	29	23	14, 17, 20, 24	2, 4, 26	43	X	X
4	7, 13, 25	1, 2, 10	24	7	X	44	X	4, 21
5	4, 26	3	25	2, 5, 18, 20, 22	X	45	2, 9, 14, 27, 30	4
6	13, 15, 18, 19, 27	1, 8, 12, 15, 21	26	1, 2, 13, 15, 26	4, 12	46	10, 13, 14	3, 4, 16
7	2, 10, 18, 21, 23	X	27	X	2, 4, 5, 16, 30	47	4, 15, 21	21, 26
8	9, 19, 23	18, 24	28	12, 13, 14, 20, 23	X	48	7, 9, 20	2, 4, 26, 29
9	14, 16, 29	2, 5, 18	29	19	1, 4	49	X	2, 4, 7, 10, 20
10	15	2, 29	30	10	1, 2, 21, 27	50	2	X
11	6	26, 29	31	1, 15	1, 5, 10			
12	4, 15	2, 7, 15	32	8	1, 3			
13	7, 13, 22	2, 4, 21	33	2, 9, 15, 20	1, 3, 18			
14	2, 13, 18, 19, 20	15	34	2, 9, 16	4			
15	7, 12, 14, 20	2, 18	35	X	4, 7, 14, 20			
16	21, 26	X	36	2, 13, 18, 19	11			
17	X	4	37	8, 10, 18, 22, 23	17			
18	8, 19, 29	3, 7, 12, 19, 20, 21	38	15, 19, 20, 21, 23, 24	12, 13, 26			
19	15	3	39	X	2, 12, 14			
20	5, 23, 26, 27	4, 10	40	24	9			

^a^ Medications listed were self-reported by subjects at the time of analysis. 1 = duloxetine, 2 = gabapentin, 3 = pregabalin, 4 = tricyclic antidepressants, 5 = fluoxetine, 6 = milnacipran, 7 = SSRI, 8 = bupropion, 9 = naltrexone, 10 = NSAIDs, 11 = topiramate, 12 = trazodone, 13 = antihistamines, 14 = stimulants, 15 = thyroid medications, 16 = testosterone, 17 = venlafaxine, 18 = benzodiazepines, 19 = statins, 20 = antihypertensives, 21 = proton pump inhibitors, 22 = aspirin, 23 = sedative/hypnotics, 24 = lamotrigine, 25 = tamoxifen, 26 = opioid analgesics, 27 = diabetes meds, 28 = no medications, 29 = muscle relaxants, 30 = anxiolytic, 31 = ropinirole; 28 = X.

**Table 5 biomedicines-11-02704-t005:** Average wavenumber and relative percentage area of IR bands deconvoluted in the spectral region from 1500 to 1700 cm^−1^.

	Wavenumber (cm^−1^)	Area (%)	Wavenumber (cm^−1^)	Area (%)	Wavenumber (cm^−1^)	Area (%)	Wavenumber (cm^−1^)	Area (%)
**FM**	* 1565 ^a^ ± 3 ^#^	36.1 ^A^ ± 18.1	1588 ^b^ ± 4	38.2 ^A^ ± 15.6	1639 ^d^ ± 5	4.7 ^C^ ± 1.6	1670 ^f^ ± 2	21.3 ^D^ ± 7.5
**LC**	-	-	1581 ^B^ ± 2	70.3 ^B^ ± 8.9	1635 ^e^ ± 7	9.9 ^C^ ± 9.2	1670 ^f^ ± 2	23.9 ^D^ ± 5.3

* Means (*n* = 40) with different lower and capital case letters are significantly different (*p* < 0.05). ^#^ Standard deviation.

**Table 6 biomedicines-11-02704-t006:** Figures of merit of OPLS-DA model for LC and FM diagnosis with one latent variable after normalization, smoothed (SG, 5 points), second derivative (SG, 7 points) and one component of orthogonal signal correction (OSC).

Figures of Merit	Calibration Set *n* = 80 Samples	Prediction Set *n* = 20 Samples
SECV/SEP	0.10	0.18
R^2^	0.98	0.96
Sensitivity (%)	100	100
Specificity (%)	100	100
Accuracy (%)	100	100

## Data Availability

The data presented in this study are available on request from the corresponding author. The data are not publicly available due to privacy concerns.

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
