# Peer review of "Metabolic Fingerprinting for the Diagnosis of Clinically Similar Long COVID and Fibromyalgia Using a Portable FT-MIR Spectroscopic Combined with Chemometrics"

_biomedicines, 2023, doi:10.3390/biomedicines11102704_

Round 1

Reviewer 1 Report

It is not clear what type of article it is.

The introduction is adequate and helps to make a meaningful prologue to the subject.

The text is well written but there are some points that need clarification. For example in figure 4a the previous text does not explain all 3 factors.

It is not explained whether the sample has been tested for normality and why this number was assigned to the sample and the control group. The percentage of men and women and whether this played a role in the research is not mentioned, but only their age.

The discussion needs expansion.

Could there be repeatability in the study?

Are there any limitations to the study? Can taking medication affect the result?

Reports want compliance as directed.

Author Response

It is not clear what type of article it is.

This is an original research manuscript.

The introduction is adequate and helps to make a meaningful prologue to the subject.

The text is well written but there are some points that need clarification. For example in figure 4a the previous text does not explain all 3 factors.

We have rewritten this part of the results section to better explain our results.

It is not explained whether the sample has been tested for normality and why this number was assigned to the sample and the control group. The percentage of men and women and whether this played a role in the research is not mentioned, but only their age.

We acknowledge the difference in gender make up amongst the LC cohort and its difference to the homogenous sample of FM subjects. The difference in the groups between men and women is presented in Table 1 We have added percentages between men and women for further clarity. Table 2 discusses that differences in age, BMI and SIQR values were not significantly different between men and women with LC. The Shapiro-Wilk normality test was conducted on the LC subjects and the test was not significant (P = 0.319) suggesting that the distribution of the sample is not significantly different from a normal distribution. This data is now presented in Table 2.

The discussion needs expansion.

The discussion has been significantly expanded.

Could there be repeatability in the study?

This is a moderate sized study that was validated and showed excellent sensitivity and specificity for discriminating the related syndromes of LC and FM. Furthermore, the deconvolution analysis on raw spectral data has shown unique signatures for all FM and LC patients.  Also, the normality data provided should attest to the fact that this is a normal distribution cohort and therefore should be reproducible.

Are there any limitations to the study? Can taking medication affect the result?

The largest limitation to the study is the small n. We have expanded upon the discussion addressing this issue. With regard to medications, there was a general similarity with regards to medication usage between the LC and FM cohort particularly with regard to use of gabapentin (12-LC, 13-FM), opioid analgesics (LC: 5, FM: 7), and selective serotonin reuptake inhibitors (SSRI) (LC: 6, FM: 4). There were some differences noted with tricyclic antidepressants being more prevalent in FM subjects than LC (LC: 3, FM: 15). and naltrexone use being more noted in LC subjects [LC: 5, FM: 1; naloxone dosage between 1 – 4.5 mg]. Whether these changes could have affected our spectral characteristics is not known at this time and will require a larger cohort for further analysis which we are embarking upon. Subsequently, to assist in removing medication effect in the analysis, we will record patient medication usage, categorize the medication by types and quantify it by amount usage. We can associate the medication usage with each metabolite biomarker by correlations or logistic regression models. In addition, with a larger cohort, we can compare subjects on similar medications such as gabapentin and eliminate confounder medications from the analyses.

Reports want compliance as directed.

NOT SURE WHAT THE REVIEWER IS ASKING HERE.

Reviewer 2 Report

The presented article aims to differentiate between Long Covid (LC) and Fibromyalgia (FM) utilizing portable FT-MIR spectroscopic techniques and chemometrics. Although the objective is made clear, the inclusion of goals within both the abstract and objectives sections presents redundancy and does not add substantive value. It could benefit from conciseness and avoidance of repetitive content to enrich the information value.

A sample size of 50 per group raises concerns regarding statistical power and generalizability. Additionally, the inclusion of a 100% female FM group versus a mixed-gender LC group presents potential biases and lacks gender representation, impacting the reliability of the results.

The study acknowledges the lack of power to determine the medication effect but fails to appropriately address or correct for the potential impact of medications on the metabolic fingerprints. A more detailed analysis or stratification of medication types and their potential influences on metabolic profiles would strengthen the validity of the study's findings.

The deconvolution of the spectral data is crucial to identifying specific metabolic markers. However, the study does not sufficiently describe the methods used for deconvolution and statistical analysis for identifying spectral markers, leaving the reliability of the identified marker at 1565 cm-1 in question.

The article fails to thoroughly address the clinical significance and relevance of the unique spectral marker at 1565 cm-1 linked to glutamate side chains. A comprehensive discussion of the relevance of this biomarker in the pathophysiology of FM is crucial for interpreting the study's clinical impact.

The study should have elucidated any statistical differences or similarities between LC and FM groups regarding age, BMI, or SIQR. The lack of such comparative analysis impacts the ability to draw meaningful conclusions from the collected data.

The paper claims a 100% accuracy and specificity, which is highly improbable in a biological setting, especially with the relatively small sample size. More detailed information on statistical analyses and statistical testing performed, p-values, confidence intervals, and potential sources of error would aid in evaluating the robustness of these findings.

The conclusion and its implications are oversimplified and overly optimistic regarding the technique’s diagnostic potential. There is a lack of critical discussion around the limitations of the study, and more conservative language is advised to avoid overinterpretation of the findings.

The identification of the unique biomarker requires validation through subsequent studies and cross-validation with other established diagnostic methods for LC and FM.

In summary, while the study presents a novel approach in distinguishing between LC and FM, it is marred by methodological limitations, lack of representativeness, inadequate detail in crucial areas, and over-assertion of its findings’ accuracy and reliability. A more cautious, detailed, and diversified approach is recommended.

Author Response

The presented article aims to differentiate between Long Covid (LC) and Fibromyalgia (FM) utilizing portable FT-MIR spectroscopic techniques and chemometrics. Although the objective is made clear, the inclusion of goals within both the abstract and objectives sections presents redundancy and does not add substantive value. It could benefit from conciseness and avoidance of repetitive content to enrich the information value.

Thank you for these comments. We have removed some redundancy from the objectives section in the introduction and left the abstract intact.

A sample size of 50 per group raises concerns regarding statistical power and generalizability. Additionally, the inclusion of a 100% female FM group versus a mixed-gender LC group presents potential biases and lacks gender representation, impacting the reliability of the results.

Thank you for these comments. We address these concerns in our expanded discussion, discussion of statistical analyses. These points are addressed in the discussion and the key points are pasted below.

The findings of this pilot study are intriguing although definitive conclusions are limited by the size of the cohort. The clinical groups were generally similar in terms of age and BMI. The FIQR is a validated surrogate marker of pain in subjects with FM. Similarly, the SIQR is a FM neutral questionnaire which asks the identical questions but does not assume that patients have FM. Patients with LC presented a wide array of clinical symptoms many of which mirror the symptom complaints of patients with FM.  The values in both groups were statistically similar suggesting that there were similar levels of pain experience in the LC and FM groups. The FM group was 100% female while the LC group had 18 males and 32 females. An obvious limitation regards the statistical power and generalizability of our data. Additionally, the inclusion of a 100% female FM group versus a mixed-gender LC group presents potential biases and lacks gender representation, impacting the reliability of the results. Importantly, we found there was no statistically significant difference between the male and female LC sub-jects with regards to age, BMI, or SIQR as noted in Table 2. Furthermore, normality tests suggest that this data has a normal distribution which strongly suggests its potential for reproducibility.

The study acknowledges the lack of power to determine the medication effect but fails to appropriately address or correct for the potential impact of medications on the metabolic fingerprints. A more detailed analysis or stratification of medication types and their potential influences on metabolic profiles would strengthen the validity of the study's findings.

Thank you again for these comments. The expanded discussion addresses several of these points. The key points of this are pasted below. Medications of the recruited patients were recorded at the time of blood collection. This pilot study was not powered to determine medication effect, however, the spectroscopy data showed that there was no obvious signal/influence from medications. With the relatively small n of 50 in each cohort, we are not able to definitively eliminate medications as a confounding variable. There was a general similarity with regards to medication usage between the LC and FM cohort particularly with regard to use of gabapentin (12-LC, 13-FM), opioid analgesics (LC: 5, FM: 7), and selective serotonin reuptake inhibitors (SSRI) (LC: 6, FM: 4). There were some differences noted with tricyclic antidepressants being more prevalent in FM subjects than LC (LC: 3, FM: 15). and naltrexone use being more noted in LC subjects [LC: 5, FM: 1; naloxone dosage between 1 – 4.5 mg]. Whether these changes could have affected our spectral characteristics is unknown at this time and will require a larger cohort for further analysis which we are embarking upon. Future studies will seek to mitigate medication effect in the analysis. We will record patient medication usage, categorize the medication by types and quantify it by amount usage. We can associate the medication usage with each metabolite biomarker by correlations or logistic regression models. In addition, with a larger cohort, we can compare subjects on similar medications such as gabapentin and eliminate confounder medications from the analyses.

The deconvolution of the spectral data is crucial to identifying specific metabolic markers. However, the study does not sufficiently describe the methods used for deconvolution and statistical analysis for identifying spectral markers, leaving the reliability of the identified marker at 1565 cm-1 in question.

We have included more details in the materials and methods section about the procedure to apply the deconvolution and the statistical analysis applied.

The article fails to thoroughly address the clinical significance and relevance of the unique spectral marker at 1565 cm-1 linked to glutamate side chains. A comprehensive discussion of the relevance of this biomarker in the pathophysiology of FM is crucial for interpreting the study's clinical impact.

We are confirming the findings of glutamate biomarker by using LC-MS/MS analysis. However, the importance of glutamate has been found to favor of a dysregulation of pain processing in the central nervous system of FM patients, particularly associated with an increase in cerebral glutamate levels. Furthermore, there is evidence to support an association between increased glutamate levels and an increase in FM symptoms (Pyke et al., 2017). In clinical studies, an increase in glutamate levels has been observed in the brains of FM patients (Holton et al., 2012; Radhakrishnan and Sluka, 2009). Furthermore, pregabalin has been observed to reduce glutamatergic activity in the insula and some subgroups of patients with FM respond to treatment with N-methyl-D-aspartate (NMDA) glutamate receptor antagonists, suggesting an increase in glutamatergic activity. Parallel to the clinical evidence, animal studies in the non-inflammatory pain model have shown increased glutamate release in the spinal and ventromedial rostral cords (Skyba et al., 2005).

The study should have elucidated any statistical differences or similarities between LC and FM groups regarding age, BMI, or SIQR. The lack of such comparative analysis impacts the ability to draw meaningful conclusions from the collected data.

There were no statistically significant differences between LC and FM groups regarding age, BMI or SIQR. Those analyses are presented in Table 3.

The paper claims a 100% accuracy and specificity, which is highly improbable in a biological setting, especially with the relatively small sample size. More detailed information on statistical analyses and statistical testing performed, p-values, confidence intervals, and potential sources of error would aid in evaluating the robustness of these findings.

We have added the Table 6 with Figures of merit for our OPLS-DA model for the calibration and validation sets to provide more detailed information. We have rewritten this section of the manuscript following the reviewer’s and editor comment.

The conclusion and its implications are oversimplified and overly optimistic regarding the technique’s diagnostic potential. There is a lack of critical discussion around the limitations of the study, and more conservative language is advised to avoid overinterpretation of the findings.

The conclusion section has been rewritten. We have included in the discussion section the limitations of the study.

The identification of the unique biomarker requires validation through subsequent studies and cross-validation with other established diagnostic methods for LC and FM.

We agree that more work needs to be done on this exciting research and we are continuing our work on identifying the nature of the biomarker by including LC-MS/MS metabolomics approaches.

In summary, while the study presents a novel approach in distinguishing between LC and FM, it is marred by methodological limitations, lack of representativeness, inadequate detail in crucial areas, and over-assertion of its findings’ accuracy and reliability. A more cautious, detailed, and diversified approach is recommended.

We have improved the manuscript based on the comments from reviewers. However, we do not agree that the manuscript is marred with methodological limitations and lack of representativeness. Although we concur that more detailed information and more caution was needed in our assertions, our data supports the excellent accuracy and specificity of the predictive model using a moderate sized number of patients. Our study includes in many cases double the number of patients in metabolomics studies using NMR and MS that have shown lower performance. We recruited a diverse pool of patients from clinics that are taking no medication or a wide array of medications and our model reproducibly grouped the different classes and the validated test group was accurately predicted.

Round 2

Reviewer 2 Report

Given a 100% accuracy and specificity, achieved, the ROC plot presented in Figure 6 is rather uninformative. 

Discuss the threats to reliability of results, such as overgeneralization and bias.

Author Response

Given a 100% accuracy and specificity, achieved, the ROC plot presented in Figure 6 is rather uninformative.

We appreciate the reviewers comments however, given that readers of the Journal and specifically this paper might be drawn to the article from varied scientific backgrounds, providing the ROC plot might aid in their understanding of that aspect of the study and the overall principle we are trying to convey. We would respectfully request to leave the Figure 6. intact.

Discuss the threats to reliability of results, such as overgeneralization and bias.

The following paragraph has been added to the Discussion section: Lines 458 - 466. 

Our pilot results must be interpreted with caution lest we run the risk of overgeneralization. First of all, although our findings are intriguing we will exercise caution rather than extrapolate our results to be indicative for all individuals affected with LC since this study is not powered to evaluate whether these signatures are characteristic of all affected with LC or only a subset or specific variant. Indeed, it is unclear at this time whether all LC is the same or if it varies in severity between variants. Secondly, enrollment bias could exist due to differences in male:female ratios in groups, medication differences between groups amongst others. Future studies with much larger sample sizes should mitigate these types of concerns